

# Freshwater carbon and nutrient cycles revealed through reconstructed population genomes

Alexandra M. Linz[1], Shaomei He[1,2], Sarah L.R. Stevens[1],
Karthik Anantharaman[1], Robin R. Rohwer[3], Rex R. Malmstrom[4],
Stefan Bertilsson[5] and Katherine D. McMahon[1,6]

[1] Department of Bacteriology, University of Wisconsin-Madison, Madison, WI, USA
[2] Department of Geoscience, University of Wisconsin-Madison, Madison, WI, USA
[3] Environmental Chemistry and Technology Program, University of Wisconsin-Madison, Madison, WI, USA
[4] Department of Energy Joint Genome Institute, Walnut Creek, CA, USA
[5] Department of Ecology and Genetics, Limnology and Science for Life Laboratory, Uppsala University, Uppsala, Sweden
[6] Department of Civil and Environmental Engineering, University of Wisconsin-Madison, Madison, WI, USA

Corresponding author
Alexandra M. Linz, amlinz@wisc.edu

## ABSTRACT

Although microbes mediate much of the biogeochemical cycling in freshwater, the categories of carbon and nutrients currently used in models of freshwater biogeochemical cycling are too broad to be relevant on a microbial scale. One way to improve these models is to incorporate microbial data. Here, we analyze both genes and genomes from three metagenomic time series and propose specific roles for microbial taxa in freshwater biogeochemical cycles. Our metagenomic time series span multiple years and originate from a eutrophic lake (Lake Mendota) and a humic lake (Trout Bog Lake) with contrasting water chemistry. Our analysis highlights the role of polyamines in the nitrogen cycle, the diversity of diazotrophs between lake types, the balance of assimilatory vs. dissimilatory sulfate reduction in freshwater, the various associations between types of phototrophy and carbon fixation, and the density and diversity of glycoside hydrolases in freshwater microbes. We also investigated aspects of central metabolism such as hydrogen metabolism, oxidative phosphorylation, methylotrophy, and sugar degradation. Finally, by analyzing the dynamics over time in nitrogen fixation genes and *Cyanobacteria* genomes, we show that the potential for nitrogen fixation is linked to specific populations in Lake Mendota. This work represents an important step towards incorporating microbial data into ecosystem models and provides a better understanding of how microbes may participate in freshwater biogeochemical cycling.

## INTRODUCTION

Lakes receive nutrients from surrounding terrestrial ecosystems (*Williamson et al., 2008*), placing lakes as "hotspots" for carbon and nutrient cycling in the landscape

(*Butman et al., 2015*). Approximately half of the carbon received by freshwater ecosystems from the terrestrial landscape is emitted as carbon dioxide (0.2 Pg C/year) or buried in sediments (0.8 Pg C/year) (*Cole et al., 2007*). Similarly, 20% of global denitrification is estimated to occur in freshwater, roughly equivalent to the amount of denitrification taking place in soils (22%) and about a third of the amount occurring in oceans (58%) (*Seitzinger et al., 2006*).

Most of this freshwater biogeochemical cycling is performed by microbial communities, yet the categories in the models and budgets used to study these cycles are too broad to incorporate microbial data. For example, carbon compounds are often classified as labile and recalcitrant (*Guillemette & Del Giorgio, 2011*), or autochthonous and allochthonous (*Jonsson et al., 2001*). While some work has been done on microbial responses to these carbon categories (*Eiler et al., 2003*; *Kritzberg et al., 2004*), using such broad categorizations masks the complexity of microbial ecophysiology. Incorporating microbially-mediated transformations of specific compounds in freshwater would significantly improve the accuracy and predictive power of biogeochemical cycling models.

However, linking microbial taxa to specific biogeochemical functions is a challenging task. Previous research has investigated substrate use by freshwater taxa using cultured isolates and microscopy fluorescence in situ hybridization coupled to microautoradiography to detect incorporation of labeled substrates in uncultured lineages (*Hahn et al., 2012*; *Salcher, Posch & Pernthaler, 2013*). While these techniques are definitive, they cannot be scaled to investigate many community members simultaneously. Other research has used scalable genomics techniques to link microbial taxa to predicted biogeochemical functions, generating hypotheses that can be tested using more targeted experiments. Sequencing data has previously been employed to great effect to analyze the distribution of functional marker genes in freshwater (*Ramachandran & Walsh, 2015*; *Peura et al., 2015*) and to predict metabolic potential in freshwater genomes (*Salcher et al., 2015*; *Eiler et al., 2016*; *Hamilton et al., 2017*; *He et al., 2017*; *Cabello-Yeves et al., 2018*).

In this research, we combined insights from both genes and genomes in three freshwater metagenomic time series to link function to taxonomy at the community level. Our metagenomic time series included multiple years of sampling for microbial DNA from two lakes in Wisconsin, USA: Lake Mendota, a large eutrophic lake, and Trout Bog Lake, a small humic lake. Mendota and Trout Bog are ideal sites for comparative time series metagenomics because of their contrasting limnological attributes and their history of extensive environmental sampling by the North Temperate Lakes–Long Term Ecological Research program (NTL–LTER, http://lter.limnology.wisc.edu) (Table 1; Table S1). They have also been the subjects of many prior efforts to document and understand freshwater bacterial community diversity and dynamics (*Shade et al., 2007*; *Linz et al., 2017*; *Hall et al., 2017*). We describe both predicted pathways in metagenome-assembled genomes (MAGs) and the distributions of functional marker genes to provide a comprehensive overview of microbially-mediated biogeochemical cycling in these two contrasting freshwater lakes.

**Table 1 Characteristics of Lake Mendota and Trout Bog Lake.**

|  | Lake Mendota | Trout Bog Epilimnion | Trout Bog Hypolimnion |
|---|---|---|---|
| Location | Madison, WI | Boulder Junction, WI |  |
| Coordinates | 43.107055, −89.411729 | 46.041172, −89.686297 |  |
| Depth of lake (m) | 25.3 | 7.9 |  |
| Surface area of lake ($km^2$) | 39.61 | 0.01 |  |
| Microbial sampling depth range (m) | 0–12 | 0–2 | 2–7 |
| Years sampled | 2008–2012 | 2007–2009 | 2007–2009 |
| Oxygenation | Oxic | Oxic | Suboxic/Anoxic |
| pH | 8.6 (0.4) | 5.0 (0.2) | 5.3 (0.2) |
| Dissolved inorganic carbon (ppm) | 41 (5) | 2.6 (2.2) | 6.9 (3.1) |
| Dissolved organic carbon (ppm) | 6.0 (6.2) | 18 (5) | 22 (6) |
| Total dissolved nitrogen (ppb) | 923 (487) | 637 (204) | 1,392 (1,031) |
| Total nitrogen (ppb) | 1,099 (521) | 831 (316) | 1,684 (1,563) |
| Total dissolved phosphorus (ppb) | 44 (51) | 15 (14) | 69 (98) |
| Total phosphorus (ppb) | 64 (52) | 32 (14) | 95 (126) |
| Sulfate (ppm) | 17 (1) | 1.2 (0.3) | 0.9 (0.7) |

**Note:**

Water from Mendota and Trout Bog was sampled weekly during the ice-free periods using an integrated water column sampler, and bacteria were collected on a 0.22 micron filter. Metagenomic sequencing was performed on DNA extracted from filters collected in 2008–2012 from Mendota and in 2007–2009 from Trout Bog. The epilimnion (upper thermal layer) was sampled in both lakes, while the hypolimnion (bottom thermal layer) was sampled only in Trout Bog. Chemistry data were collected by NTL–LTER from depth discrete samples taken from zero to four meters for Mendota, zero meters for the Trout Bog epilimnion, and three and seven meters for the Trout Bog hypolimnion. Values reported here are the means of all measurements in the sampling time span for each lake, with standard deviations reported in parentheses.

Throughout this paper, we highlight several functional categories with particularly interesting results. We discuss differences in the identity and diversity of potential nitrogen fixing bacteria in Trout Bog vs. Mendota, as well as the high prevalence of genes related to polyamines, which are proposed to be an important component of the dissolved organic nitrogen pool. We observed that assimilatory sulfate reduction pathways were encoded more frequently than dissimilatory sulfate reduction pathways, in contrast to what is thought to be the case in marine systems. We split the broader category of primary production into different types of phototrophy, including photosynthesis performed by *Cyanobacteria*, green sulfur bacteria, and aerobic anoxygenic phototrophs, and analyzed their associated carbon fixation pathways (when present). Using annotations of carbohydrate-active enzymes, we compared the potential for complex carbon degradation and describe significant differences in the coding density and diversity of these encoded enzymes between lakes. To compare more basic properties of freshwater microbes, we assessed differences between lakes in central microbial metabolisms such as hydrogen metabolism, oxidative phosphorylation, methylotrophy, and degradation of low molecular weight carbon. Finally, we show how trends over time in the abundances of both nitrogen fixation marker genes and *Cyanobacteria* MAGs likely encoding nitrogen fixation were highly correlated, demonstrating how genomic data can reveal dynamics in both functions and taxa.

## METHODS

### Sampling

Samples were collected from Lake Mendota and Trout Bog Lake as previously described (*Bendall et al., 2016*). Briefly, integrated samples of the water column were collected during the ice-free periods of 2007–2009 in Trout Bog and 2008–2012 in Mendota. In Mendota, the top 12 m of the water column were sampled, approximating the epilimnion (upper, oxygenated, and warm thermal layer). The epilimnion and hypolimnion (bottom, anoxic, and cold thermal layer) of Trout Bog were sampled separately at depths determined by measuring temperature and dissolved oxygen concentrations. The sampling depths were most often zero to two meters for the epilimnion and two to seven meters for the hypolimnion. DNA was collected by filtering 150 mL of the integrated water samples through 0.2-µm pore size polyethersulfone Supor filters (Pall Corp., Port Washington, NY, USA). Filters were stored at −80 °C until extraction using the FastDNA Spin Kit (MP Biomedicals, Burlingame, CA, USA) with minor modifications (*Shade et al., 2007*).

### Sequencing

As previously described (*Bendall et al., 2016*; *Roux et al., 2017*), metagenomic sequencing was performed by the Department of Energy Joint Genome Institute (DOE JGI) (Walnut Creek, CA, USA). A total of 94 samples collected over 5 years were sequenced for Mendota, while 47 metagenomes collected over 3 years were sequenced for each layer in Trout Bog (Table S2). Samples were sequenced on the Illumina HiSeq 2500 platform (Illumina, San Diego, CA, USA), except for four libraries (two from each layer of Trout Bog) that were sequenced using the Illumina TruSeq protocol on the Illumina GAIIx platform; all samples were sequenced using paired ends with read lengths of 150 base pairs (Data S1). Paired-end sequencing reads were merged with FLASH v1.0.3 with a mismatch value of less than 0.25 and a minimum of 10 overlapping bases (*Magooc & Salzberg, 2011*). 16S rRNA gene amplicon sequencing was also performed on samples collected with the same method over the same time periods. These datasets are available under DOE JGI project IDs 1078703 and 1018581 for Trout Bog and Mendota, respectively. Samples from Trout Bog were sequenced on the 454 GS FLX-Titanium platform (Roche, Branford, CT, USA) targeting the V8 hypervariable region (primer 1392R: ACGGGCGGTGTGTRC) (*Engelbrektson et al., 2010*), and sequences were trimmed to 324 base pairs using VSEARCH (v2.3.4) (*Rognes et al., 2016*). Samples from Mendota were sequenced on an Illumina MiSeq, and the V4 region was targeted using paired-end sequencing (primers 525F: GTGCCAG CMGCCGCGGTAA and 806R: GGACTACHVGGGTWTCTAAT) (*Caporaso et al., 2012*). Both datasets were trimmed based on alignment quality and chimera checking using mothur v.1.39.5 (*Schloss et al., 2009*). Unclustered, unique sequences were classified using a custom database of freshwater 16S rRNA gene sequences (*Newton et al., 2011*) and the Greengenes database (*DeSantis et al., 2006*) with the classification pipeline TaxAss (*Rohwer et al., 2018*).

## Assembly and binning

To recover MAGs, metagenomic reads from the same sampling sites (Mendota's epilimnion, Trout Bog's epilimnion, and Trout Bog's hypolimnion) were pooled (Table S2) and then assembled as previously described (Bendall et al., 2016; Roux et al., 2017). In metagenomes from Trout Bog, this assembly was performed using SOAPdenovo2 at various k-mer sizes (Luo et al., 2012), and the resulting contigs were combined using Minimus (Sommer et al., 2007). In Mendota, merged reads were assembled using Ray v2.2.0 with a single k-mer size (Boisvert et al., 2012). Contigs from the combined assemblies were binned using MetaBAT ("-veryspecific" settings, minimum bin size of 20 kb, and minimum contig size of 2.5 kb) (Kang et al., 2015), and reads from individual metagenomes were mapped to the assembled contigs using the Burrows–Wheeler Aligner ($\geq$95% sequence identity, $n = 0.05$) (Li & Durbin, 2010), which allowed time-series resolved binning (Table S2). DOE JGI's Integrated Microbial Genome (IMG) database tool (https://img.jgi.doe.gov/mer/) (Markowitz et al., 2012) was used for gene prediction and annotation. Annotated MAGs can be retrieved directly from the IMG database and JGI's Genome Portal using the IMG Genome ID provided (also known as IMG Taxon ID). MAG completeness and contamination/redundancy was estimated based on the presence of a core set of genes with CheckM (Rinke et al., 2013; Parks et al., 2015), and MAGs were taxonomically classified using Phylosift (Darling et al., 2014) or the phylogeny-based "guilt by association" method (Hamilton et al., 2017). As recommended by Bowers et al. (2017), only MAGs that were at least approximately 50% complete with less than 10% estimated contamination/redundancy (meeting the MIMARKS definition of a medium or high quality MAG) (Bowers et al., 2017) were included in the study.

A total of 193 medium to high quality bacterial MAGs were recovered from the three combined time series metagenomes in Trout Bog and Mendota: 99 from Mendota, 31 from Trout Bog's epilimnion, and 63 from Trout Bog's hypolimnion (Data S2). These population genomes ranged in estimated completeness from 50 to 99% based on CheckM estimates. Several MAGs from Trout Bog's epilimnion and hypolimnion appeared to belong to the same population based on average nucleotide identities greater than 99% calculated using DOE JGI's ANI calculator (Data S3) (Varghese et al., 2015). This is likely because assembly and binning were carried out separately for each thermal layer, even though some populations were present throughout the water column.

## Functional marker gene analysis

To analyze functional marker genes in the unassembled, unpooled metagenomes, we used a curated database of reference protein sequences (Data S4) (Anantharaman et al., 2016) and identified open reading frames (ORFs) in our unassembled metagenomic time series using Prodigal (Hyatt et al., 2010). This analysis was conducted on merged reads. The protein sequences and ORFs were compared using BLASTx (Camacho et al., 2009) with a cutoff of 30% identity. Read abundance was normalized by metagenome size for plotting. We chose to perform this analysis because gene content in unassembled metagenomes is likely more quantitative and more representative of the entire microbial

community than gene content in the MAGs, due to limitations of assembly and binning algorithms.

These comparisons were run between the epilimnia of Trout Bog and Mendota, and between the epilimnion and hypolimnion of Trout Bog. We did not compare Mendota's epilimnion to Trout Bog's hypolimnion, as the multitude of factors differing between these two sites make this comparison illogical. We aggregated marker genes by function (as several marker genes from a phylogenetic range were included in the database for each type of function) and tested for significant differences in distribution between lakes and layers using a Wilcoxon rank sum test in R with a Bonferroni correction for multiple pairwise testing.

### Pathway prediction

Pathways were analyzed by exporting IMG's functional annotations for the MAGs, including KEGG, COG, PFAM, and TIGRFAM annotations, and mapping to pathways in the KEGG and MetaCyc databases as previously described (He et al., 2017). To score presence, a pathway needed at least 50% of the required enzymes encoded by genes in a MAG, and if there were steps unique to a pathway, at least one gene encoding each unique step. Putative pathway presence was aggregated by lake and phylum in order to link potential functions identified in the metagenomes to taxonomic groups that may perform those functions in each lake. Glycoside hydrolases were identified using dbCAN2's implementation of HMMER (Zhang et al., 2018). Nitrogen usage in amino acids was calculated by taking the average number of nitrogen atoms in translated ORF sequences across each MAG.

Data formatting and plotting was performed in R (R Core Team, 2017) using the following packages: ggplot2 (Wickham, 2009.), cowplot (Wilke, 2017), reshape2 (Wickham, 2007), and APE (Paradis, Claude & Strimmer, 2004). The datasets, scripts, and intermediate files used to predict pathway presence and absence are available at https://github.com/McMahonLab/MAGstravaganza. Any future updates or refinements to this dataset will be available at this link.

## RESULTS AND DISCUSSION

### Community functional marker gene analysis

Due to the contrasting water chemistry of Mendota and Trout Bog (Table 1; Table S1), we expected that microbial metabolisms would differ between lakes, and that these differences would be reflected in metagenomic gene content. To assess the potential for differing microbial metabolisms by lake, we tested whether functional marker genes identified in the unassembled merged metagenomic reads appeared more frequently in one lake or layer compared to the others. Many functional markers were found to be significantly more abundant in specific sites; more will be reported in each of the following sections (Fig. 1; Table S3). The recovered MAGs represent a diverse set of genomes assigned to taxonomic groups typically observed in freshwater (Fig. S2).

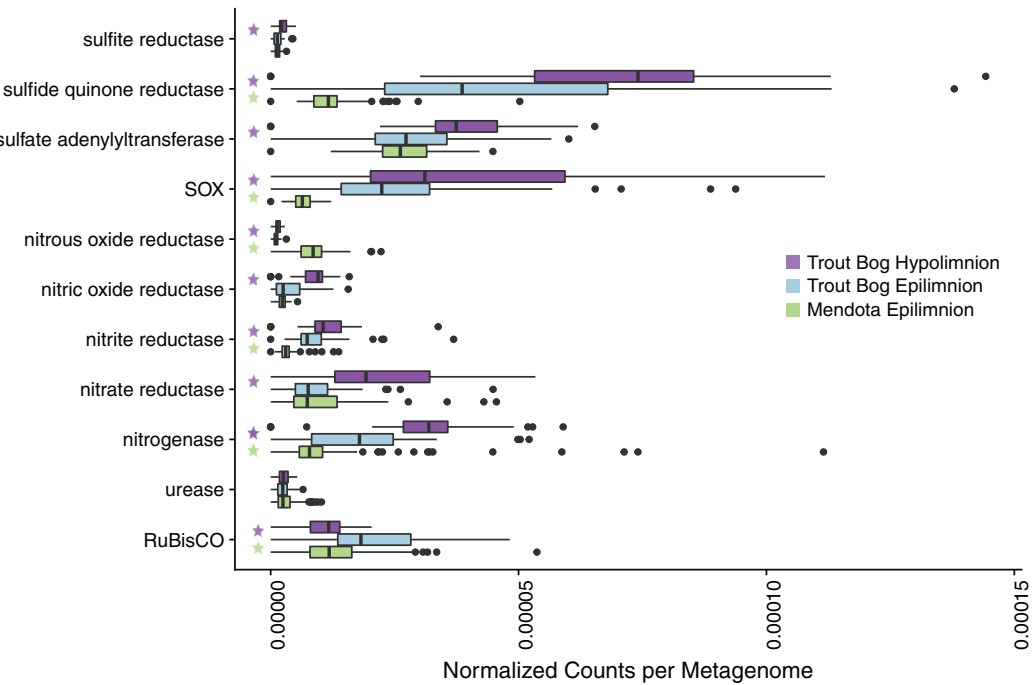

**Figure 1 Analysis of marker gene abundances reveals differences between lakes and layers.** To assess potential differences in microbial metabolisms in our study sites, we predicted open reading frames in unassembled metagenomes using Prodigal and compared the resulting ORFs to a custom database of metabolic marker genes using BLAST. In these boxplots, significant differences in numbers of gene hits between sites were tested using a pairwise Wilcoxon rank sum test with a Bonferroni correction; significance was considered to be $p < 0.05$. A total of 94 metagenomes were tested for Mendota, while 47 metagenomes were tested in each layer of Trout Bog. Significant differences between the Trout Bog and Mendota epilimnia and between the Trout Bog epilimnion and hypolimnion are indicated by a green or a purple star, respectively. Significant differences between the Trout Bog hypolimnion and the Mendota epilimnion were not tested, as the large number of variables differing in these sites makes the comparison less informative. This analysis revealed differences in the number of marker genes observed by lake for many metabolic processes involved in carbon, nitrogen, and sulfur cycling. $p$-values of markers described in Fig. 1 and elsewhere in the text are reported in Table S3.

## Overview of the MAGs dataset

To identify the phylogenetic affiliations of the microbes carrying marker genes and the co-occurrences of key marker genes within the same population genomes, we used MAGs from each metagenomic time series to predict metabolic pathways based on genomic content. To assess the diversity of our MAGs, we constructed an approximate maximum likelihood tree of all the MAGs in FastTree (*Price, Dehal & Arkin, 2010*) using whole genome alignments (Fig. S1). The tree is not intended to infer detailed evolutionary history, but to provide an overall picture of similarity between genomes. MAGs recovered are a diverse set of genomes assigned to taxa typically observed in freshwater (Fig. S2).

We also compared 16S rRNA gene amplicon sequencing data from the same timeframe as the metagenomes to confirm that the microbial community composition for these lakes and years was not "abnormal" compared to previous published studies (Fig. S3).

The observed taxonomic compositions were consistent with other 16S-based studies carried out on these lakes (*Linz et al., 2017*; *Hall et al., 2017*) and with freshwater bacterial community compositions in general (*Newton et al., 2011*).

## Nitrogen cycling

Nitrogen availability is an important factor structuring freshwater microbial communities. It is often a determining factor in a lake's trophic status and a risk factor for the development of toxic cyanobacterial blooms (*Smith, 2003*; *Beversdorf, Miller & McMahon, 2013*). Because of the significance of nitrogen in freshwater, we analyzed nitrogen-related marker genes and identified MAGs containing characteristic nitrogen cycling pathways. We discovered significant differences in the abundances of marker genes, along with differences in phylogenetic affiliations of the MAGs containing these pathways.

Genes encoding nitrogenase, the key enzyme in nitrogen fixation, were observed most frequently in metagenomes from Trout Bog's hypolimnion, followed by Trout Bog's epilimnion, and lastly by Mendota's epilimnion (Fig. 1; Table S3). We analyzed MAGs predicted to fix nitrogen and found differences in the identities of putative diazotrophs between the two ecosystems (Fig. 2; Fig. S1 and Data S5). In Mendota, two-thirds of MAGs encoding the nitrogen fixation pathway were classified as *Cyanobacteria*, while the other third was assigned to *Betaproteobacteria* and *Gammaproteobacteria*. Although not all *Cyanobacteria* fix nitrogen, previous studies of nitrogen fixation in Mendota have reported a strong correlation between this pathway and the cyanobacterium affiliated with *Aphanizomenon* (*Beversdorf, Miller & McMahon, 2013*). MAGs containing genes encoding nitrogen fixation were more phylogenetically diverse in Trout Bog and included *Deltaproteobacteria*, *Gammaproteobacteria*, *Epsilonproteobacteria*, *Acidobacteria*, *Verrucomicrobia*, *Chlorobi*, and *Bacteroidetes*. The higher diversity of diazotrophs in Trout Bog compared to Mendota suggests that nitrogen fixation may be a more advantageous trait in humic lakes than in eutrophic lakes.

We noted a high frequency of genes related to polyamine biosynthesis and degradation in our MAGs. We found that 94% of MAGs encoded pathways for polyamine synthesis, and 87% encoded pathways for polyamine degradation. These pathways were predicted in diverse MAGs from both lakes, including *Actinobacteria* as previously observed (*Ghylin et al., 2014*; *Hamilton et al., 2017*). While there is some evidence for the importance of polyamines in aquatic systems (*Mou et al., 2011*), the ecological roles of these compounds in freshwater are not fully resolved. Polyamines are known to play a critical but poorly understood role in bacterial metabolism (*Igarashi & Kashiwagi, 1999*), and the exchange of these nitrogen compounds between populations may be a factor structuring freshwater microbial communities. Polyamines can also result from the decomposition of amino acids, so higher trophic levels such as fish or zooplankton may represent an additional polyamine source (*Al Bulushi et al., 2009*). The frequent appearance of polyamine-related pathways in our MAGs lends support to the hypothesis that these compounds are important but largely unrecognized parts of the dissolved organic nitrogen and carbon pool in freshwater.

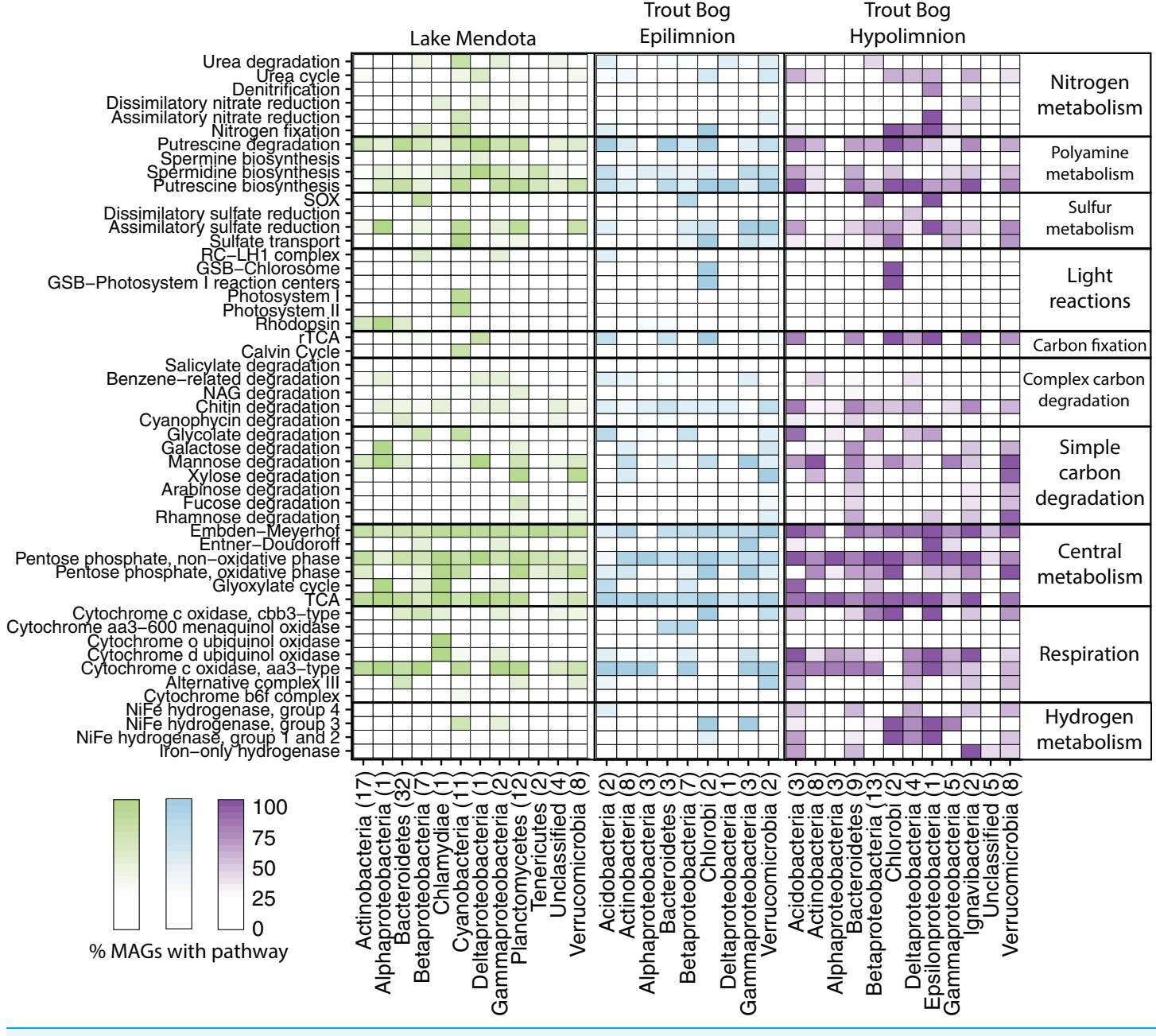

**Figure 2  Metabolisms in Mendota and Trout Bog.** A pathway was considered present when at least 50% of enzymes in a pathway were encoded in the genome and all enzymes unique to or required for the pathway were present. Putative pathway presence was aggregated by lake and phylum. This analysis can link potential functions identified in the metagenomes to taxonomic groups that may perform those functions. For example, MAGs that putatively fix carbon also likely fix nitrogen in both lakes. Similarly, putative degradation pathways for rhamnose, fucose, and galactose were frequently encoded within the same MAGs. *Proteobacteria* was split into classes due to the high diversity of this phylum. The number of MAGs assigned to each phylum is indicated in parentheses after the phylum name. Data for each genome can be found in Data S6.

We analyzed genes for denitrification, including reductases for nitrous oxide, nitrite, and nitrate. Denitrification genes were observed most frequently in Trout Bog's hypolimnion, with the exception of nitrous oxide reductase, which was found more frequently in Mendota. Genes encoding urease were not identified more frequently in any

site. Denitrification and urea degradation pathways were predicted in similar proportions of MAGs from both lakes.

## Sulfur cycling

Sulfur is another essential element in freshwater that is cycled between oxidized and reduced forms by microbes. Our marker gene analysis demonstrated that genes encoding sulfide:quinone reductase (for sulfide oxidation) and the sox pathway (for thiosulfate oxidation) were significantly more abundant in Trout Bog compared to Mendota, with no significant differences between the layers of Trout Bog (Fig. 1; Table S3). Genes encoding sulfite reductases were the least abundant sulfur cycling marker genes in all sites. Dissimilatory sulfite reductase was observed only in MAGs from Trout Bog, especially those classified as *Chlorobiales*. Because this enzyme is thought to operate in reverse in green sulfur-oxidizing phototrophs such as *Chlorobiales* (*Holkenbrink et al., 2011*), this may indicate an oxidation process rather than a reductive sulfur pathway. Sulfur oxidation pathways were observed in MAGs classified as *Betaproteobacteria* from both lakes and *Epsilonproteobacteria* in Trout Bog's hypolimnion. Assimilatory sulfate reduction was overall the most common sulfur-related pathway identified in the MAGs (Fig. 2; Data S5).

Assimilatory sulfate reduction was observed more frequently than dissimilatory sulfate reduction; this suggests that sulfate is more commonly used for biosynthesis, while reduced forms of sulfur are used as electron donors for energy mobilization in these populations. This is in contrast to marine systems, where sulfate reduction holds a central role as an energy source for organotrophic energy acquisition (*Bowles et al., 2014*), although sulfate reduction could also be occurring in Mendota's hypolimnion.

## Phototrophy

Primary production (the coupling of photosynthesis and carbon fixation) is a critical component of the freshwater carbon cycle. To identify differences in routes of primary production between freshwater environments, we compared marker genes for carbon fixation across sites. Ribulose-1,5-bisphosphate carboxylase/oxygenase (RuBisCO), the marker gene for carbon fixation via the Calvin–Benson–Bassham (CBB) pathway, was most frequently observed in Trout Bog's epilimnion (Fig. 1; Table S3).

We assessed the MAGs for photoautotrophy, expecting to find differences between our two study sites based on the observed contrasts in the functional marker gene analysis (Fig. 2; Data S5). In Mendota, the majority of MAGs encoding phototrophic pathways were classified as *Cyanobacteria*. These MAGs contained genes encoding enzymes in the CBB pathway. In Trout Bog, most MAGs encoding phototrophy were classified as *Chlorobium clathratiforme*, a species of *Chlorobiales* widespread in humic lakes (*Karhunen et al., 2013*). The *Chlorobiales* MAGs in Trout Bog contained genes encoding citrate lyase and other key enzymes in the reductive tricarboxylic acid (TCA) cycle, an alternative carbon fixation method commonly found in green sulfur bacteria such as *Chlorobi* (*Kanao et al., 2002*; *Tang & Blankenship, 2010*). Although we found genes annotated as the RuBisCO large subunit (*rbcL*) in some of the *Chlorobiales* MAGs, the reductive TCA cycle is the only carbon fixation pathway known to be active in cultured

representatives of *Chlorobiales*. Homologs of *rbcL* have been previously identified in isolates of *Chlorobium*, and were associated with sulfur metabolism and oxidative stress (*Hanson & Tabita, 2001*). Given this information, it seems likely that this *rbcL* homolog encodes a function other than carbon fixation in our *Chlorobiales* MAGs. MAGs affiliated with *Cyanobacteria* in Mendota and *Chlorobi* in Trout Bog also possessed genes encoding diazotrophy, providing a link between carbon and nitrogen fixation. As both *Chlorobi* and *Cyanobacteria* are often abundant members of freshwater communities (*Eiler & Bertilsson, 2004*; *Peura et al., 2012*), their fixation capabilities may be relevant even at the ecosystem scale.

The potential for photoheterotrophy via the aerobic anoxygenic phototrophic (AAP) pathway was identified in several MAGs from all lake environments, especially from epilimnia, based on the presence of genes annotated as *pufABCLMX, puhA*, and *pucAB* encoding the core reaction center RC-LH1 (*Martinez-Garcia et al., 2012b*). *Betaproteobacteria* and *Gammaproteobacteria*, particularly MAGs classified as *Burkholderiales* (including PnecC, LD28, and *Zwartia alpina*), most often contained these genes, although they were not broadly shared across the phylum (Fig. 2). As AAP has previously been associated with freshwater *Proteobacteria* (*Martinez-Garcia et al., 2012b*), these results are not surprising. However, an *Acidobacteria* MAG from the Trout Bog epilimnion also contained genes suggesting AAP, which to our knowledge has not previously been found in this phylum.

Another form of photoheterotrophy previously identified in freshwater is the use of light-activated proteins such as rhodopsins (*Martinez-Garcia et al., 2012b*). We observed genes encoding rhodopsins in MAGs from each lake environment, but more frequently in *Actinobacteria* and *Bacteroidetes* MAGs from Mendota (Fig. 2). Trout Bog, especially the hypolimnion, harbored fewer and less diverse MAGs encoding rhodopsins than those from Mendota.

## Glycoside hydrolases

Degradation of high-complexity, recalcitrant carbon compounds requires specialized enzymes, but wide availability of these carbon compounds can make complex carbon degradation an advantageous trait. One way to predict the ability to degrade high-complexity carbon in microbial populations is by identifying genes annotated as glycoside hydrolases (GHs), which encode enzymes that break the glycosidic bonds found in complex carbohydrates. However, it is important to keep in mind that GHs can also play structural roles in microbial cells in addition to the degradation of complex carbon substrates (*Henrissat & Davies, 1997*). A previous study of *Verrucomicrobia* MAGs from our dataset found that the profiles of GHs differed between Mendota and Trout Bog, potentially reflecting the differences in available carbon sources (*He et al., 2017*). We expanded this analysis of GHs to all of the MAGs in our dataset to identify differences in how populations from our two study sites degrade complex carbohydrates.

We calculated the coding density of GHs, defined as the percentage of coding regions in a MAG annotated as a GH, to identify differences in carbon metabolism between MAGs from different lake environments (Fig. 3; Data S6). Our GH coding density metric

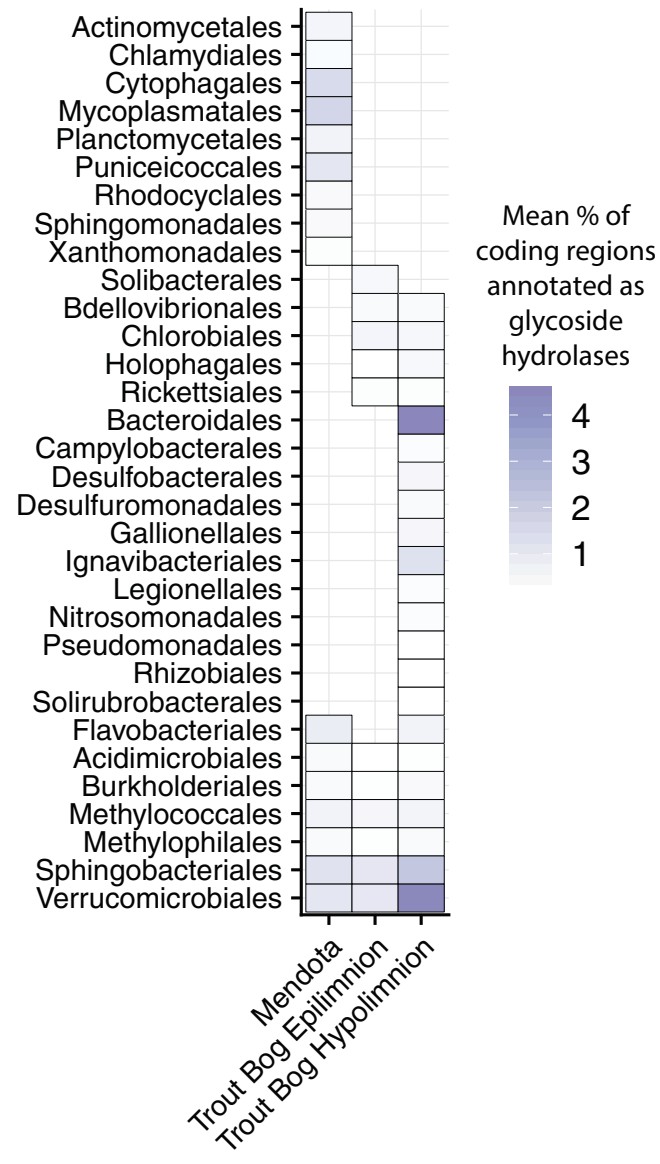

**Figure 3 Glycoside hydrolase content in the MAGs.** Annotations of GHs were used as an indication of complex carbon degradation. Genes potentially encoding GHs were identified and assigned CAZyme annotations using dbCAN2. GH coding density was calculated for each MAG and averaged by order and lake. While a few orders contained genes encoding glycoside hydrolases in all three sites, many orders were unique to each site. The orders with the highest coding densities were all found in the Trout Bog hypolimnion. Glycoside hydrolase diversity, an indicator of the range of substrates an organism can degrade, was significantly correlated with coding density ($r^2 = 0.92$, $p < 2.2 \times 10^{-16}$). *Proteobacteria* was split into classes due to the high diversity of this phylum.

was significantly correlated with the diversity of GHs identified ($r^2 = 0.92$, $p < 2.2 \times 10^{-16}$), which is an indicator of the number of substrates an organism can utilize. The MAGs with the highest GH coding densities were classified as *Bacteroidales, Ignavibacteriales, Sphingobacteriales*, and *Verrucomicrobiales* from Trout Bog's hypolimnion. Two of these orders, *Sphingobacteriales* and *Verrucomicrobiales*, also contained MAGs with high GH coding densities in Mendota and Trout Bog's epilimnion. There were several

additional orders with high GH coding density that were unique to Mendota, including *Mycoplasmatales (Tenericutes)*, *Cytophagales (Bacteroidetes)*, *Planctomycetales (Planctomycetes)*, and *Puniceicoccales (Verrucomicrobia)*. Members of *Verrucomicrobia* have been previously identified as potential polysaccharide degraders in freshwater, although our coding densities for this phylum are higher than previously reported (*Martinez-Garcia et al., 2012a*). This may be due to differences in trophic status between our lakes and those previously studied, or it may be that MAGs capture more pan-genomic content than isolate or single amplified genomes. In concordance with their ability to hydrolytically degrade biopolymers to sugars, MAGs with high GH coding densities also contained putative degradation pathways for a variety of sugars (Fig. 2). The increased diversity of these genes found in Trout Bog's hypolimnion compared to the other study sites suggests differing diversity and complexity of the available organic carbon.

## Central metabolism and simple carbon degradation

Freshwater microbes are exposed to a great variety of low-complexity carbon sources such as carbohydrates, carboxylic acids, and single-carbon (C1) compounds. The central metabolic pathways shared by most living cells are often an entry point for the least complex carbon compounds. Therefore, the specific routing of central metabolism predicted in our MAGs may reveal how low complexity carbon compounds are used within freshwater populations.

We investigated the types of cytochrome oxidases encoded in our MAGs to compare oxidative phosphorylation between lakes and layers (Fig. 2; Data S5). Cytochrome c oxidases, both aa3- and cbb3-type, were widespread in all three lake environments and frequently co-occurred within MAGs. aa3-type cytochromes are associated with high oxygen concentrations, while cbb3-type cytochromes are associated with low oxygen concentrations (*Gong et al., 2018*). The presence of genes encoding both types suggests the flexibility to operate under a range of oxygen concentrations.

Similarly, hydrogen metabolism can influence and be influenced by other aspects of nutrient usage. Iron-only hydrogenases were found primarily in MAGs from Trout Bog's hypolimnion (Fig. 2; Table S3), consistent with their previously identified presence in anaerobic, often fermentative bacteria (*Peters et al., 2015*). Group 3 [Ni–Fe] hydrogenases were identified in MAGs belonging to *Cyanobacteria* and *Chlorobiales* in both lakes. This finding is consistent with the proposed function of Group 3d, which is to remove excess electrons produced by photosynthesis (*Peters et al., 2015*).

Low molecular weight carbohydrates may be derived either from autochthonous (such as algae) or allochthonous (such as terrestrial plants) sources (*Giroldo, Augusto & Vieira, 2005*; *Ramanan et al., 2016*). The pathway for mannose degradation was encoded in many MAGs from all three sites (Fig. 2; Data S5). Predicted pathways for rhamnose, fucose, and galactose degradation were often found within the same MAGs (including members of *Planctomycetes* and *Verrucomicrobia* from Mendota, and members of *Bacteroidetes, Ignavibacteria*, and *Verrucomicrobia* from Trout Bog). Xylose is a common freshwater sugar which has already been proposed as a potential carbon source for streamlined *Actinobacteria* (*Ghylin et al., 2014*). We confirmed this in our MAGs and also

identified *Bacteroidetes*, *Planctomycetes*, and *Verrucomicrobia* from Mendota and *Bacteroidetes* and *Verrucomicrobia* from Trout Bog as additional potential xylose degraders. Genes for the degradation of glycolate, an acid produced by algae and consumed by heterotrophic bacteria (*Paver & Kent, 2010*), were identified in *Cyanobacteria* and *Betaproteobacteria* MAGs from Mendota and in *Acidobacteria, Verrucomicrobia, Alpha-, Beta-, Gamma-,* and *Epsilonproteobacteria* MAGs from Trout Bog. The pathways predicted in our MAGs may inform us about which low molecular weight compounds are important carbon substrates in freshwater.

Methylotrophy, the ability to grow solely on C1 compounds such as methane or methanol, was predicted in MAGs from both Trout Bog and Mendota. Putative pathways for methanol and methylamine degradation were found in MAGs classified as *Methylophilales* (now merged with *Nitrosomonadales*; *Boden, Hutt & Rae, 2017*), while *Methylococcales* MAGs were potential methane degraders based on the presence of genes encoding methane monooxygenase. *Methylococcales* MAGs from Trout Bog also encoded the pathway for nitrogen fixation, consistent with reports of nitrogen fixation in cultured isolates of this taxon (*Bowman, Sly & Stackebrandt, 1995*). Methylotrophy in cultured freshwater isolates from *Methylococcales* and *Nitrosomonadales* is well-documented (*Kalyuzhnaya et al., 2011*; *Salcher et al., 2015*). We also found predicted pathways for methanol degradation in MAGs classified as *Burkholderiales* and *Rhizobiales* in Trout Bog. Methylotrophy has been identified in members of *Rhizobiales*, such as *Methylobacterium* and *Methylocystaceae*, and in *Burkholderiales*, including *Methylibium* (*Auman et al., 2000*; *Chistoserdova et al., 2003*; *Kane et al., 2007*). Our MAGs may represent populations related to these known methylotrophs.

## Using MAGs to track population abundances over time

Because our metagenomes comprise a time series, we can investigate potential changes in function over time using our MAGs and functional marker genes. We analyzed nitrogen fixation over time in *Cyanobacteria*, known to be highly variable over time in Mendota. We found that in each year, one *Cyanobacteria* MAG was substantially more abundant (based on read coverage) than the rest; this single MAG was plotted for each year in Mendota (Figs. 4A–4E). We compared read coverage-based abundance of the dominant *Cyanobacteria* MAG to the normalized number of BLAST hits in the metagenomes from abundant functional marker genes encoding nitrogenase subunits (TIGR1282 (*nifD*), TIGR1286 (*nifK* specific for molybdenum–iron nitrogenase), and TIGR1287 (*nifH*, common among different types of nitrogenases)) (Figs. 4F–4J). As expected, we detected significant correlations ($p < 0.05$) between MAG abundance and nitrogen fixation marker genes in 2008, 2011, and 2012. In these years, the dominant *Cyanobacteria* MAGs were predicted to fix nitrogen based on gene content, while the dominant MAGs in 2009 and 2010 were not predicted to fix nitrogen. In agreement with this, the number of hits for the nitrogenase marker genes were an order of magnitude lower in 2009 and 2010 compared to 2008 and 2012. While genome incompleteness precludes us from concluding that the potential for nitrogen fixation in Mendota was lower in 2009 and 2010 because the dominant *Cyanobacteria* populations were not diazotrophic,

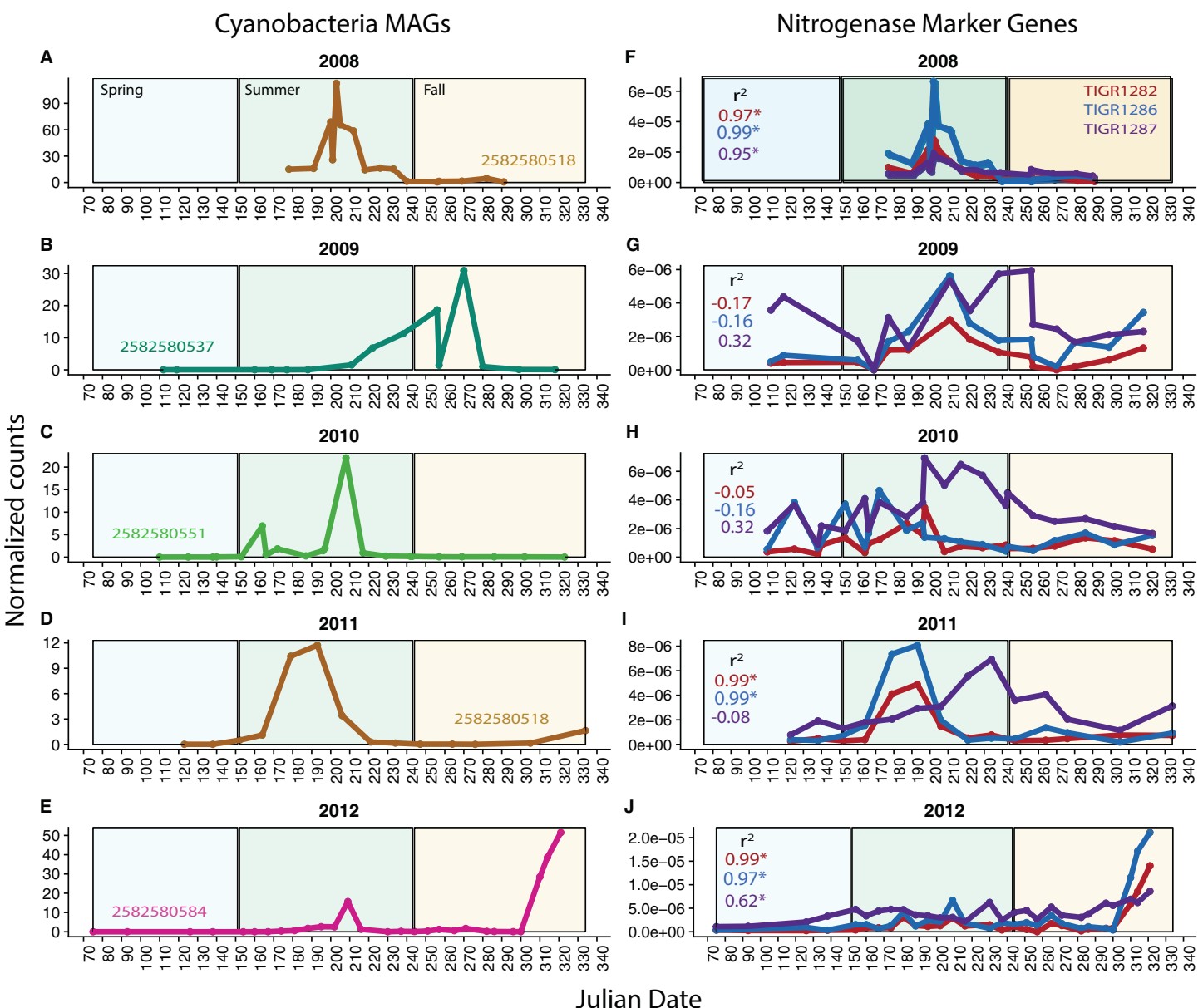

**Figure 4 Cyanobacteria and nitrogen fixation over time.** To investigate potential functional changes over time in Mendota, we compared the abundances of *Cyanobacteria* MAGs (approximated using read coverage normalized by genome length) to the abundances of nitrogen fixation marker genes (approximated using the number of BLAST hits to metagenomes normalized by metagenome size). Only the most abundant *Cyanobacteria* MAG is shown for each year (A–E) because a single MAG was much more abundant than the rest in each year. The marker genes used were TIGR1282, TIGR1286, and TIGR1287, encoding subunits of Mo–Fe nitrogenase, as these were the most frequently observed nitrogenase markers in the Mendota metagenomes (F–J). Significantly correlated trends over time were observed between the MAGs and the nitrogenase marker genes in 2008, 2011, and 2012. In years where there was no significant correlation, the dominant MAG did not contain genes indicative of the nitrogen fixation pathway. This suggests that *Cyanobacteria* dynamics may be linked to the potential for nitrogen fixation in Mendota.

it does suggest a strong link between *Cyanobacteria* dynamics and nitrogen fixation in this ecosystem (*Beversdorf, Miller & McMahon, 2013*). This could also have important implications for cyanotoxin production, since nitrogen stress has been linked to toxin production (*Beversdorf et al., 2015*).

## CONCLUSIONS

Our analysis of functional marker genes indicated potentially significant differences in microbial biogeochemical cycling between Mendota's epilimnion, Trout Bog's epilimnion, and Trout Bog's hypolimnion. We next used MAGs from multi-year metagenomic time series to propose specific roles in freshwater biogeochemical cycles for microbial taxa. In the nitrogen cycle, we predicted many pathways for the degradation and biosynthesis of polyamines, consistent with their hypothesized role in the dissolved organic nitrogen pool. We observed an association between nitrogen fixation and *Cyanobacteria* in Mendota, but observed a greater diversity of putative diazotrophs in Trout Bog. Assimilatory sulfate reduction pathways were predicted more frequently that dissimilatory sulfate reduction pathways, suggesting a bias towards using sulfate for biosynthesis. We identified several types of phototrophy, which in some but not all genomes co-occurred with carbon fixation via the Calvin Cycle or the reductive TCA cycle. We found the greatest diversity and density of GHs in MAGs from Trout Bog's hypolimnion, suggesting a greater potential to degrade recalcitrant carbon in this region. Our combination of functional marker gene analysis and MAG pathway prediction provides insight into the complex metabolisms underpinning freshwater communities and how microbial processes scale to ecosystem functions.

We anticipate that this dataset will be a valuable community resource for other freshwater microbial ecologists to mine and incorporate into comparative studies across lakes around the world. As such, all data is publicly available at https://github.com/McMahonLab/MAGstravaganza. The results of this study can be used to guide efforts to build microbially-resolved models of freshwater carbon and nutrient cycles with better predictive power.

## ACKNOWLEDGEMENTS

We thank the North Temperate Lakes—Long Term Ecological Research Program and Lake Mendota Microbial Observatory field crews, UW-Trout Lake Station, the UW Center for Limnology, and the Global Lakes Ecological Observatory Network for field and logistical support. We acknowledge efforts by many McMahon laboratory undergraduate students and technicians whose work has been related to sample collection and DNA extraction. We thank Emily Stanley and Joshua Hamilton for insightful comments on an early draft of this manuscript. Finally, we personally thank the individual program directors and leadership at the National Science Foundation for their commitment to continued support of long-term ecological research.

### Funding

This research was supported by the U.S. Department of Energy Joint Genome Institute through the Community Sequencing Program (CSP 394). The work conducted by the U.S. Department of Energy Joint Genome Institute, a DOE Office of Science User Facility, is supported by the Office of Science of the U.S. Department of Energy under

Contract No. DE-AC02-05CH11231. Katherine D. McMahon received funding from the United States National Science Foundation Microbial Observatories program (MCB-0702395), the Long Term Ecological Research Program (NTL–LTER DEB-1440297), and an INSPIRE award (DEB-1344254). Alexandra M. Linz was supported by a pre-doctoral fellowship provided by the University of Wisconsin–Madison Department of Bacteriology and by the National Science Foundation Graduate Research Fellowship Program under grant no. DGE-1256259 during this research. This material is also based upon work supported by the National Institute of Food and Agriculture, U.S. Department of Agriculture (Hatch Project 1002996). The funders had no role in study design, data collection and analysis, decision to publish, or preparation of the manuscript.

### Grant Disclosures

The following grant information was disclosed by the authors:
U.S. Department of Energy Joint Genome Institute through the Community Sequencing Program: CSP 394.
U.S. Department of Energy Joint Genome Institute.
Office of Science of the U.S.
Department of Energy under Contract No. DE-AC02-05CH11231.
United States National Science Foundation Microbial Observatories program: MCB-0702395.
Long Term Ecological Research Program: NTL–LTER DEB-1440297.
INSPIRE award: DEB-1344254.
University of Wisconsin–Madison Department of Bacteriology and by the National Science Foundation Graduate Research Fellowship Program: DGE-1256259.
National Institute of Food and Agriculture.
U.S. Department of Agriculture: Hatch Project 1002996.

### Competing Interests

The authors declare that they have no competing interests.

### Author Contributions

- Alexandra M. Linz conceived and designed the experiments, analyzed the data, prepared figures and/or tables, authored or reviewed drafts of the paper, approved the final draft.
- Shaomei He analyzed the data, contributed reagents/materials/analysis tools, authored or reviewed drafts of the paper, approved the final draft.
- Sarah L.R. Stevens analyzed the data, contributed reagents/materials/analysis tools, authored or reviewed drafts of the paper, approved the final draft.
- Karthik Anantharaman analyzed the data, contributed reagents/materials/analysis tools, authored or reviewed drafts of the paper, approved the final draft.
- Robin R. Rohwer analyzed the data, contributed reagents/materials/analysis tools, authored or reviewed drafts of the paper, approved the final draft.

- Rex R. Malmstrom conceived and designed the experiments, authored or reviewed drafts of the paper, approved the final draft.
- Stefan Bertilsson conceived and designed the experiments, authored or reviewed drafts of the paper, approved the final draft.
- Katherine D. McMahon conceived and designed the experiments, prepared figures and/or tables, authored or reviewed drafts of the paper, approved the final draft.

## DNA Deposition

The following information was supplied regarding the deposition of DNA sequences:

Metagenomes, pooled metagenome assemblies, and metagenome-assembled genomes (MAGs) described here are accessible through the Integrated Microbial Genomes (IMG) database. IMG Genome IDs for these many sequences can be found at https://github.com/McMahonLab/MAGstravaganza (also included as Supplemental Documents).

## Data Availability

McMahon Lab Github–MAGstravaganza

https://github.com/McMahonLab/MAGstravaganza

## Supplemental Information

Supplemental information for this article can be found online at http://dx.doi.org/10.7717/peerj.6075#supplemental-information.

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
