# Peer review of "Freshwater carbon and nutrient cycles revealed through reconstructed population genomes"

_PeerJ, doi:10.7717/peerj.6075_

## Round 0.1 · original submission · Minor Revisions

Your paper has a good potential but requires minor revisions. Please modify the manuscript in accord with reviewer's comments.

·

Basic reporting

No comment.

Experimental design

No comment.

Validity of the findings

Linz et al. present additional analyses of 3 time-series metagenomic datasets, which were collected over several years from 2 fractions of a humic lake, Trout Bog (3 years) and 1 fraction of eutrophic Lake Mendota (5 years). They provide 1) gene-centric analyses of the metagenomes (pooled by fraction) examining the differential distribution of a set of functional marker genes, and 2) population-genome-centric analyses of metagenome-assembled-genome bins (MAGs). With the marker-gene analyses they infer differences in multiple metabolic pathways among the lakes and water column fractions. They then use the MAGs to associate the identified processes with specific taxa, finding that certain metabolic processes appear to co-occur within taxa, and that common processes across sites are associated with (and attributed to) differing phylogenetic groups. This manuscript is very well written and organized and materials and scripts have been provided to make it reproducible. The following two main concerns regarding their analyses and interpretations should be addressed:

1) 'Community Functional Marker Gene Analysis' Lines 179-194
• A technical concern with the marker-gene analysis is that the authors are using merged reads (Lines 143-147) for which they predict ORFs and use blast to assign function. Previously the authors stated that the merged reads range from 150-290bp (Lines 110-111). The authors do not indicate the read lengths used for sequencing, this should be added to the relevant supplementary information, or if all are using the same read lengths (2x150?), this can be added to the text. Even if these are all using the same read-lengths, please test that the distribution of merged read lengths does not differ significantly across sample groups. I am concerned that a skewed distribution toward shorter merged read lengths would result in the underprediction/failure to detect marker genes, particularly longer ones. If your method already controls for this, please add a statement explaining how.

• Even once this is resolved, I am confused by the reporting of the marker gene analysis. It is unclear why the authors used LEfSE to identify significant differences in gene 'abundance' among sites (Line 149), yet Figure 1 employs and presents results from pairwise Wilcoxon rank sum tests. I understand that in the second case the authors are combining multiple genes together to simplify the presentation; however in the case of citrate lyase, Figure 1 indicates that the two comparisons are significant by this combined test, but the LEfSE results for aclA and aclB report "not_significant" for both (TB layers, ML/TB epi) comparisons. Please explain this discrepancy. Do your conclusions vary based on which results you are reporting? Additionally, please add to Supplemental Data S2 a notation so that it is clear which markers are combined to generate the categories on the y-axis in Figure 1. Should you choose to display only the LEfSE results, a form similar to that used in Figure 2 might be better showing the actual genes and their groupings.

2) Overall, the remaining descriptions of the metabolic pathways identified within the MAGs were good but could benefit from additional interpretation. There were several paragraphs where it felt that the authors were describing at length the results, which are nicely displayed in Figure 2, but I was left wondering 'why do I care?' Please place these discussions in the environmental context. A particular example is the paragraph on the cytochrome oxidases. To summarize, my recommendation is to revisit these paragraphs and confirm the conclusions were strongly stated and the linkages to differing chemistries across sites made effectively.

Additional comments

Additional points to address:
In 'How Representative are the MAGs?' (Lines 195-223) and 'Using MAGs to track population abundances over time' (Lines 469-491):
• The authors make a qualitative comparison at the phylum level across the datasets and methods indicating that the assignments "largely" agree. What is the point to this analysis? If anything this analysis underscores the bias associated with all methods of describing microbial community composition and undermines the implicit assumption across the manuscript that the MAGs are representative of the community, and therefore are accountable for the observed differences in marker genes. Please summarize why this comparison matters and how it supports the conclusions reached in this manuscript. Please explain explicitly at the end of this paragraph what the main takeaway from this analysis is and how it is related to the rest of the observations. Or remove.
• If the authors would like to rely on this similarity to ascribe the observed marker-gene abundances to the MAGs with similar pathways, this could be better achieved by making a direct comparison of the merged reads (marker genes) identified with those encoded in the MAGs in order to quantify which are actually associated with the assembled MAGs and what proportion remained unassembled. I believe this is what the authors are trying to do with the final analysis of the paper, linking nitrogenase marker genes to the dominant cyanobacterial MAG, however in that analysis it is also unclear what exactly is being measured and compared. It seems that the authors are correlating the abundances of nitrogenase genes with the abundances of the Cyanobacteria MAG. Why then not make a direct comparison of sequences? Lines 485-6, state "it is possible that other diazotrophic populations were more abundant, or that the nitrogenase subunits were derived from populations that did not assemble into MAGs." The authors should have the data available to actually test this statement. Please explain if there is some reason why this analysis could not be done, otherwise I think it would strengthen these observations if it is possible to directly attribute the nitrogenase counts to the MAGs rather than making the indirect link presented in Figure 4.

• Line 213 - "...consistent with a higher likelihood of recovering MAGs from the most abundant populations in the community." Please provide either a reference or analysis to support this statement or remove it. It has been demonstrated that high-abundance populations can in fact be underrepresented in assembly (PMID 26033198 and others) due to technical issues with assembling strain variants. Rather, it is often the opposite, where low abundance, low-diversity populations are the most amenable to assembly from metagenomic data. Finally as the authors also noticed, some taxa are not identified via 16S rRNA amplicon sequencing despite their representation in the metagenome. (for example PMID 26083755 and others)

• Line 221 - 16S rRNA amplicon sequencing not just "16S" please.

• Lines 244-246. Why does the observed diversity suggest horizontal gene transfer? Are these taxa not typically associated with nitrogen fixation? Is this diversity particularly surprising for some reason? Does analysis of the MAGs suggest these particular genes have been horizontally transferred?

• Line 326: "The co-occurrence of fixation pathways in these pathways in these populations are especially interesting given their relatively high abundance in their respective lakes." Why is this especially interesting? Please elaborate.

• Line 328-336: The discussion of the presence of rbcL in Chlorobium, is lacking some connections. What is the genomic context of rbcL in non-Chlorobiales and in the Chlorobium isolates. Is the context described for these new MAGs totally different from these other observations?

• Line 344- "Unexpectedly, an Acidobacteria MAG from the Trout Bog epilimnion also contained genes suggesting aerobic anoyxgenic phototrophy." And? Why is this specifically called out - please explain why this is novel or interesting and worth the mention.

• Lines 352-398: Complex carbon degradation:
GH are necessary for complex-carbon degradation, but are also employed for structural purposes within an organism, this should be acknowledged, as the abundance and diversity of GH identified here are being used as a proxy for complex carbon degradation, and this may not be accurate.
In lines 338-391, the authors indicate that the relative numbers of GH identified may not be relevant, but then proceed to talk primarily about the most frequently identified GH families (lines 394-6; legend to Figure 3b).
Finally, the authors identify GH % values approaching 5%. This seems high (PMID 22536372, Figure 4). Both this study and that of He et al. (presumably using similar methods) identified these high percentages. If these are really accurate percentages, this seems like it would be a big deal and should be called out. I wonder if there is some technical artifact related to assembly that might be inflating this value. If so, I don't think that it changes the comparisons of these MAGs to each other, but to other genomes using different methods it might not be an appropriate comparison. Please either a) acknowledge this large discrepancy and comment on its possible source (biological or technical) or b) show from the literature how this is not an unusually high value. Regarding this analysis in Figure 3 B, C, D please add additional categories to the bar chart. Surely there are others that might be biologically interesting to include here. The large 'other' bar does not provide much information.

• Lines 493-494. These results have shown significant differences in marker gene abundances indicating POTENTIAL differences in microbial nutrient cycling. Please adjust.

Other comments on Supplementary files:

S1- Please check that the data for the following samples is correctly entered, as these lines differ in format from the other samples. HZIF, HZHX

S2 - Please add titles indicating which data is shown in panels A and B so that this is clear even without the legend. What is y-axis showing in panel B? This is unclear.

S6 - This table is incredibly sparse and not especially useful in this format. Please consider melting in R and retaining only those comparisons with non-zero ANI values. As a reduced 3-column file, this would be sortable and more readily usable for people who will make use of this excellent MAG dataset in the future.

Reviewer 2 ·

Basic reporting

While the English was clear and professional, I found the writing style at times repetitive. For example, three out of four paragraphs in the ‘Nitrogen cycling’ section begin with ‘To identify…’. I also think that sections can be shortened (e.g. ‘Community Functional Marker Gene Analysis’, l. 178-194 which seems largely redundant with the methods section).
The introduction is rather short. It should be extended to include hypotheses about the data and to expand on the motivation of the present study. For example, the manuscript could be set up as a study of methodology (how does marker gene analysis perform compared to MAGs) or of comparing lakes with different environmental conditions (Mendota vs. Trout Bog) or in terms of the time series. I found that many of the results presented were not clearly connected to the introduction, making it difficult for me to evaluate their importance.
Results and Discussion are merged into one section, which does, as far as I know, not conform to PeerJ structural standards. I think it would help the manuscript if results and discussion would be teased apart because it would allow the authors to highlight main results and discuss their relevance in detail more.
The figures were clear, relevant and of high quality.

Experimental design

The research is original and within the scope of the journal. As mentioned above, I found the research question not to be clearly defined. The research was conducted to a high technical and ethical standard and the methods are described well. However, I would encourage the author to explain the experimental design in more detail (e.g. how many samples per year in the time series).

Validity of the findings

The data are robust and the analysis statistically sound. However, I think overall it could be presented in less detail, allowing more room for fewer results to be discussed in detail and linked to hypotheses. The conclusions are a bit muddy, because a clear research question is lacking.

Additional comments

The authors investigate 141 metagenomics datasets sampled as a time series from two lakes utilizing both functional marker gene and metagenomic binning approaches. As expected, the results obtained from this impressive dataset are complex and not easily summarized. However, I do feel that the presented manuscript fell a bit short in aiding the reader to understand these complexities and evaluate their importance in light of the existing literature. Specifically, I found that the manuscript in its present form lacked hypotheses about the work presented. As a consequence, the presented results come across a bit like a long list of findings, but their relevance is hard to gage.

- Title: The title is a bit vague. What is connected exactly and what is the nature of these connections.
- L. 44-50: The flow of this paragraph is a bit strange.
- L. 115: Do primers for the different V regions have comparative biases? Otherwise, it might be questionable to compare 16S taxonomies between lakes.
- L. 121-124: I don’t understand the sentence. Please clarify.
- L. 132: Please clarify how many datasets were pooled? Based on which criteria?
- L. 155: The sentence is redundant with respect to the previous paragraphs.
- L. 180-194: This paragraph should be significantly shortened or incorporated into the methods section.
- L. 208ff.: This part should likely be moved to the methods section.
- L. 226: Vague. What specific differences and why?
- L. 233: I don’t think O2 is the explanation here, lots of organisms have the ability to fix N2 in the presence of oxygen, including cyanos.
- L. 246: Interesting. But could it not also have evolved independently? Is there support for HGT (e.g. nitrogen fixation genes are very conserved between different taxa).
- L. 273-285: I don’t understand the point of this paragraph. I recommend removing it.
- In general in the results/Discussion sections: background and motivation to study a specific pathway is given after a result is introduced. Maybe expand on the background info in the introduction.
- L. 271: The hypothesis is not mentioned before.
- L. 328-336: This paragraph adds little to the results. I recommend removing.
- A table summarizing the results for functional marker genes and MAGs might be useful.
- L. 351-361: This section is about glycoside hydrolases. So I’d remove the opening paragraph, as it’s not helpful. And change the title to reflect what this section is about.
- L. 469: This is the first time this is mentioned. Set the result up in the introduction with background information and a hypothesis.

---

## Round 0.2 · Minor Revisions

I have one final additional comment. In discussing methylotrophy, it comes through that you are a little surprised with finding it in Rhizobiales and Burkholderiales. Note that some of the most well studied methylotrophs, such as Methylobacterium, along with alphaproteobacterial methanotrophs, members of Methylocystaceae, all belong to Rhizobiales. Methylotrophy has also long been known in Burkholderiales, (and related Rhodocyclales), represented, for example, by members of the genus Methylibium. Thus, I suggest you modify the manuscript accordingly.

---

## Round 0.3 · accepted · Accept

Congratulations, your paper is accepted!

#